# Combining Nanopore Sequencing with Recombinase Polymerase Amplification Enables Identification of Dinoflagellates from the *Alexandrium* Genus, Providing a Rapid, Field Deployable Tool

**DOI:** 10.3390/toxins15060372

**Published:** 2023-06-01

**Authors:** Robert G. Hatfield, David Ryder, Annabel M. Tidy, David M. Hartnell, Karl J. Dean, Frederico M. Batista

**Affiliations:** Centre for Environment Fisheries and Aquaculture Science, Weymouth DT48UB, UKanna.tidy@cefas.gov.uk (A.M.T.); frederico.batista@cefas.gov.uk (F.M.B.)

**Keywords:** *Alexandrium*, nanopore sequencing, RPA, harmful algal bloom, HABs, saxitoxin, paralytic shellfish poisoning, in-field sequencing, food safety, aquaculture, VolTRAX

## Abstract

The armoured dinoflagellate *Alexandrium* can be found throughout many of the world’s temperate and tropical marine environments. The genus has been studied extensively since approximately half of its members produce a family of potent neurotoxins, collectively called saxitoxin. These compounds represent a significant threat to animal and environmental health. Moreover, the consumption of bivalve molluscs contaminated with saxitoxin poses a threat to human health. The identification of *Alexandrium* cells collected from sea water samples using light microscopy can provide early warnings of a toxic event, giving harvesters and competent authorities time to implement measures that safeguard consumers. However, this method cannot reliably resolve *Alexandrium* to a species level and, therefore, is unable to differentiate between toxic and non-toxic variants. The assay outlined in this study uses a quick recombinase polymerase amplification and nanopore sequencing method to first target and amplify a 500 bp fragment of the ribosomal RNA large subunit and then sequence the amplicon so that individual species from the *Alexandrium* genus can be resolved. The analytical sensitivity and specificity of the assay was assessed using seawater samples spiked with different *Alexandrium* species. When using a 0.22 µm membrane to capture and resuspend cells, the assay was consistently able to identify a single cell of *A. minutum* in 50 mL of seawater. Phylogenetic analysis showed the assay could identify the *A. catenella*, *A. minutum*, *A. tamutum*, *A. tamarense*, *A. pacificum*, and *A. ostenfeldii* species from environmental samples, with just the alignment of the reads being sufficient to provide accurate, real-time species identification. By using sequencing data to qualify when the toxic *A. catenella* species was present, it was possible to improve the correlation between cell counts and shellfish toxicity from r = 0.386 to r = 0.769 (*p* ≤ 0.05). Furthermore, a McNemar’s paired test performed on qualitative data highlighted no statistical differences between samples confirmed positive or negative for toxic species of *Alexandrium* by both phylogenetic analysis and real time alignment with the presence or absence of toxins in shellfish. The assay was designed to be deployed in the field for the purposes of in situ testing, which required the development of custom tools and state-of-the-art automation. The assay is rapid and resilient to matrix inhibition, making it suitable as a potential alternative detection method or a complementary one, especially when applying regulatory controls.

## 1. Introduction

Phytoplankton represent one of Earth’s most vital resources for the persistence of life, through carbon fixation, the release of oxygen, and food provision for organisms higher up in the food chain [1]. Yet, in certain circumstances, driven by fluctuations in environmental conditions, and, at times, exacerbated by anthropogenic activities, these organisms can proliferate and reach high enough densities to cause disruption within the surrounding environment [2,3]. Such events are commonly referred to as harmful algal blooms (HABs) [4]. HAB events are enhanced when the causative organism(s) have the capabilities to produce harmful metabolites such as biotoxins [5,6]. Biotoxins are known to bioaccumulate within filter feeding organisms such as bivalve molluscs, where they present a significant hazard to other organisms, including humans who consume contaminated shellfish [2].

The majority of marine HAB events are caused by protists belonging to the phylum *Dinoflagellata,* of which the genus *Alexandrium* are, arguably, the most hazardous [6]. The currently agreed taxonomy organizes *Alexandrium* into 30 morphologically and genetically defined species, many of which are considered toxic [7,8,9,10]. The genus can produce three different families of toxins: saxitoxins, spirolides, and gonyautoxins [11]. The saxitoxin family consists of potent neurotoxins, causing disruption to voltage-gated sodium channels and, therefore, regulation of the nervous system [12]. Exposure to even low levels of the compound results in a potentially lethal syndrome known as paralytic shellfish poisoning (PSP), which is why the known analogues of saxitoxin (STX) are colloquially referred to as paralytic shellfish toxins (PST) and are listed in schedule one of the chemical weapons conventions [13,14]. The extreme toxicity of saxitoxin means that even relatively low densities of *Alexandrium* can present a human health risk, justifying widespread routine monitoring of shellfish toxins as well as an early warning system to detect *Alexandrium* cells within shellfish harvesting areas [15,16,17]. In the UK, shellfish testing for biotoxins is achieved using analytical chemistry techniques, which replaced bioassays over a decade ago [18,19]. Since their adoption, these methods have been refined and optimized to improve sensitivity and throughput, whilst new methods have been developed to improve accuracy and include more analogues of saxitoxin [20,21].

Conversely, sea water analysis is performed using the Utermöhl light microscopy method, a technique that has barely changed in nearly a century [22,23]. The longevity of its application highlights its efficacy, but it is time consuming and challenging to identify organisms to species level, often resulting in poor correlations with toxicity [24,25]. This is primarily due to the occurrence of both toxic and non-toxic species within the sampling waters [26]. To address this, a variety of alternatives have been developed using a wide array of technologies, many of which have made use of nucleic acid detection techniques, usually using the polymerase chain reaction (PCR) to ensure adequate sensitivity [27]. Adoption of these assays has not been widespread, as they usually target a specific strain or species, meaning that multiple assays would need to be run for each variant that posed a hazard. An alternative is to use a PCR assay with a lower degree of specificity, which could detect any *Alexandrium* spp., but would have no discriminatory power [28]. This could be used as a screening method, with positive samples forwarded for speciation by, for example, DNA sequencing [29]. One of the drawbacks of using PCR is the requirement for samples to be free from contaminants, for example, humic acids, which can inhibit the reaction and provide false negatives [30]. DNA extraction kits that purify DNA to a sufficient quality that is suitable for PCR can be expensive and laborious, a significant drawback when considering incorporation into a routine monitoring program. Recombinase polymerase amplification (RPA) is an alternative that has several benefits: it is isothermal (37–41 °C), rapid (<40 min), and not as susceptible to matrix inhibition [31]. Although not widely used for routine testing, sequencing is commonly used for research purposes. Until recently, the most widely adopted technology for high throughput sequencing consisted of sequencing relatively short fragments of DNA in parallel, using clonal amplification and sequencing by synthesis, commonly referred to as next generation sequencing [32,33,34,35]. However, the capability to sequence long, single strands of DNA in parallel and in real-time marked a technological breakthrough, which some refer to as third-generation sequencing [36]. This capability is predominantly provided by Pacific biosciences (Pacbio) and Oxford Nanopore Technologies (ONT), with nanopore sequencing technology being more widely adopted. This is, to some degree, due to the release of the MinION, a small, low-cost sequencing platform that generates data in real-time using transmembrane protein channels [37,38]. Although individual read accuracies are lower on ONT platforms, it is possible to use bioinformatics to generate consensus sequences that provide higher accuracies (>99.99%) [39,40]. Due to the small physical size and low cost of the MinION, it is possible for it to be used outside of the laboratory environment [41,42].

The assay outlined in this study aimed to identify *Alexandrium* down to species level by using Nanopore sequencing to analyse a ~500 bp fragment generated by RPA from the D2 region of the rRNA large subunit (LSU). The assay was designed to be field deployable by incorporating the use of cutting-edge automation and developing custom tools. This novel application of technology has notable benefits over comparable assays and has the potential for further development into rapid screening tools for industry or potential applications to statutory monitoring.

## 2. Results and Discussion

### 2.1. Method Development

#### 2.1.1. Primer Design

Candidate primers were designed and chosen based on an alignment of reference sequences (shown in Figure 1). This allowed the generation of primers with minimal degeneracies that would be inclusive of all listed organisms. This was with the exception of *Alexandrium leei*, for which three variants were required in the forward primer set (shown in Figure 1). As this species does not produce toxins that have been shown to have negatively impacted human health, it was decided not to risk the efficacy of the primers by including degeneracies in their design [43]. The assay would likely not have been effective when targeting the species *Alexandrium leei*; however, no testing was performed to verify efficacy.

Forward and reverse primers varying in length between 30 and 36 bp, targeting the same part of the ribosomal rRNA cassette, were tested using a DNA extract from a reference culture of *A. catenella*. This required 49 different reactions to screen for the best performing pair, with the primers shown in Table 1. The results from each reaction are shown in Figure 2. Reactions for the six best performing combinations were repeated on a smaller gel run, with reaction 27 found to perform the best. As such, all onward experiments and analyses used the primers from reaction 27 (specifically, Fwd31 and Rev33 primer sets).

#### 2.1.2. RPA Performance

Assessing RPA performance was challenging, due to all the samples being found to be positive based on fluorescence testing using a Qubit. Gel and capillary electrophoresis showed a smear between 100 and 700 bp in many samples. The sequencing of these samples identified the smears to be mostly either homopolymer or tandem repeats. Smears tended to occur in samples with low or undetectable levels of target DNA. In such cases, the concentration of DNA based on fluorescence testing would often exceed that calculated for positive samples. Figure 3 shows the position of the targeted amplicons, relative to non-specific amplification, and compares the accuracy of estimating peak areas from capillary and gel electrophoresis. Based on comparison of gel and sequencing results, it was found to be possible to identify positive samples using gel electrophoresis by looking for a narrow band at approximately 500 bp. This could, therefore, be used as a screening process for deciding if an RPA reaction was successful and if an environmental sample warranted sequencing. The development of a lateral flow or TwistDX exo assay would be ideally suited to this application, providing rapid screening and being better suited to the point of contact testing [44,45,46]. Alternatively, an enzyme linked oligonucleotide assay could be developed for use in a centralized laboratory for higher throughput [47]. Moreover, if the assay were to be adopted, it is likely that sequencing may not be necessary if probes targeting problem species were generated, therefore reducing cost and increasing throughput.

### 2.2. Method Optimization

Table 2 displays the results from testing the assay performance prior to sequencing, specifically, testing the sensitivity of the assay after using different sample concentration methods and testing the assay against both live cells and a mock community.

#### 2.2.1. Sensitivity

The seawater volume used in testing was 50 mL. Therefore, a single cell observed represented a detection limit of 20 cells per liter. This volume was selected, as it was the same as is used in the low cell density Utermöhl chamber, which was part of the existing light microscopy method described in the introduction. The RPA assay was consistently able to amplify genetic material from a single *Alexandrium* cell when using either direct transfer to 400 µL distilled water prior to lysis or by the filtration method. It did, however, fail to amplify one of the samples spiked with a single cell following the use of centrifugation for concentration. It is suspected that the cell was lost when the supernatant was removed, indicating that this technique was less robust than filtration. Filtration was also advantageous, as it was more conducive with field deployment. Overall, these results indicated that the assay’s sensitivity was comparable to light microscopy.

#### 2.2.2. Specificity and Selectivity

Alignment of reference sequences in the region targeted by the primers indicated that they should amplify all species of *Alexandrium* listed, with the notable exception of *A.leei*. This same approach also indicated the potential for the amplification of species from the toxic genera of the dinoflagellate *Ostreopsis*. However, lab tests using a reference strain of *Ostreopsis ovata* did not result in amplification.

The mock community consisted of ten combined phytoplankton reference cultures. Successful RPA reactions of these mock communities when spiked with *Alexandrium minutum* indicated that the assays were resilient to matrices associated with PCR inhibition. Conversely, there was no amplification in either the un-spiked mock community or the natural samples when *Alexandrium* was absent. This indicated that the assay was not amplifying any other genera present in the water samples tested. BLASTn analysis of the consensus sequences generated from all the positive environmental samples only aligned with *Alexandrium* references in the NCBI database, indicating that none of the other species present were amplified. Furthermore, sequencing of environmental samples that had no clear band at ~500 bp did not result in raw reads or consensus sequences aligning to anything in the NCBI database, and inspection of these highlighted this, as suspected, to be homopolymer or tandem repeats.

In total, five *Alexandrium* reference cultures of difference species (*A. catenella*, *A. minutum*, *A. tamutum*, *A. pacificum*, and *A. tamarense*) resulted in successful amplifications and phylogenetic identifications. *A. ostenfeldii* was also detected in the environmental study, as described in Section 2.6, with its presence being corroborated by the sequencing of a long-range amplicon generated from the same sample as part of a different study [29].

Direct comparison of seawater samples spiked with *Alexandrium* fixed with Lugol’s solution and unfixed indicated that the assay was effective in both cases. Additionally, Lugol’s fixed samples were found to consistently result in successful amplification, even after long term storage (>5 years at 5 °C), indicating that the assay could be retrospectively applied to samples in storage.

### 2.3. Real Time Alignment and Phylogenetic Tree Analysis

#### 2.3.1. Real Time Alignment

Figure 4 shows the result of aligning sequences from a mock community sample spiked with *A. minutum* against references for *Alexandrium* spp. in real time using MinKNOW 5.4.3. The reference list consisted of 50 sequences with 23 distinct species. In this sample, 94.9% of the reads were found to correctly align with the *A. minutum*, with data available in real time, throughout sequencing. Unfiltered, real-time alignments for other species tested resulted in similar proportions of reads aligning against the correct reference (*A. tamarense* 95.3%, *A. catenella* 99.8%, *A. pacificum* 99.4%, and *A. tamutum* 98.7%). As the alignments occurred in real-time, it was possible to have an indication of what species were present within minutes of starting sequencing, dependent on the sequencing rate. This indicated that the generation of consensus sequences and phylogenetic analysis may not be necessary with further testing, and real-time alignment may be adequate for species identification. Furthermore, as nanopore sequencing improves in accuracy, it is likely that there will be less need for the generation of consensus sequences in the future [48,49]. This will be further assisted by the adoption of adaptive sampling and adaptive sequencing, which allow for online sample enrichment [50,51].

#### 2.3.2. Phylogenetic Tree Analysis

The phylogenetic trees generated for this study used 50 *Alexandrium* reference sequences covering 23 species, as well as *Prorocentrum minimum* as an outgroup. The same sequences were used as were originally included in the manuscript by T. Shikata et al. [43]. An example of the phylogenetic tree is shown in Figure 5, which highlights sequences generated by the assay. Although some branches of the tree have low bootstrap scores (<75), all of the branches separating species were well supported (>95), except for the branches between *A. margalefeii* and *A. lee*; *A. hiranoi* and *A. taylori*; as well as the branches between *A. tamutum*, *A. inseutum*, and *A. minutum*.

### 2.4. Library Preparation and Multiplexing

#### 2.4.1. Oxford Nanopore Field Sequencing Kit

The field sequencing kit ref: SQK-LRK001 (LRK) was the simplest to use and fastest kit, taking only 15 min to prepare a cleaned sample for nanopore sequencing and taking under two hours to generate data from a raw sample. However, the kit only allowed for the analysis of single samples, and this lack of multiplexing made it the most expensive with low throughput.

#### 2.4.2. Oxford Nanopore Rapid PCR Barcode Kit

The rapid PCR barcode kit (RPB) facilitated the analysis of up to 12 samples concurrently. The time taken to perform RPB, clean up, quantitation, and combination into a library was approximately 4 h and was not conducive to being used in challenging situations that may be encountered in the field. Moreover, the additional PCR and requirements to quantify and compile a library using quantitation was hindered by the challenges in quantifying RPA amplicons.

#### 2.4.3. VolTRAX Multiplexing Kit

Using the VolTRAX multiplexing kit (VMK), the VolTRAX automated the barcoding and library preparation for up to ten samples. The process of multiplexing the cleaned RPA amplicons took about two hours, of which about 45 min required analyst input. With practice, the process became relatively easy to use and would be suitable for many field deployment situations. Figure 6 shows the workflow and expected turnaround time for the complete assay when using the VMK. Note: the quoted time is for collecting sufficient data to receive an initial indication from real time alignments. Consensus sequences will require more sequencing. The VMK kit required the use of a disposable cartridge, which, when combined with support for a lower number of barcodes, increased the cost when compared with the rapid PCR barcode kit.

### 2.5. Field Deployable Tools

#### 2.5.1. In Field Cell Lysis

Comparison between the Osci-lyser and the MP biological Fastprep 24 resulted in average recoveries of 1.29 and 1.34 ng/µL and 3.2 and 4.1% relative standard deviations, respectively, suggesting only small differences in the recovery of DNA from *Alexandrium* cells between the two methods. This was further assessed using a light microscope, which highlighted no intact cells after 20 s of lysis using the Osci-lyser.

#### 2.5.2. Bento Lab

The Bento Lab, a portable molecular workstation, was used for all thermal incubation, including the RPA and library preparations. It was also used for the PCR thermocycling requirements associated with barcoding with the RPB kit and the centrifugation of the samples.

#### 2.5.3. The XavION Sequencer

The XavION could perform base calling approximately ten times faster than the MinION mk1C. The ability to do this using a device that consumed a low amount of power was critical for field deployment. Although the Xavier AGX module’s power consumption was dependent on system load, it could be throttled to limit consumption to 10 W, 15 W, or 30 W. When unthrottled, the XavION consumed 80–100 watts while simultaneously performing high accuracy base calling and aligning sequences against a reference list. With the computer module throttled to 30 watts, using six of its eight cores (the setting found to be most appropriate for in field sequencing), the XavION consumed a total of ~50 watts. Therefore, when using this setting, the 185 Wh internal battery pack provided approximately three hours of sequencing before needing to be connected to an external power source. The device required considerable IT support to ensure the ONT software updates were compatible, as it was not a supported device, and settings needed to be customized. The ARM chipset used in the Xavier AGX unit was also not supported by as many software repositories as x86–64 devices, making the installation of bioinformatic tools more difficult, which meant that many of these had to be used via cloud computing. Software support is expected to improve over time, with the increasing use of the ARM chipsets in high performance computing.

### 2.6. Environmental Samples

In total, five species of *Alexandrium* were detected in the environmental study, with an overlapping succession throughout the sampling period. This agrees with previous studies of these waters [8,9,52,53]. Negative samples that were sequenced either generated no consensus sequences or were found to generate base repeats, supporting the hypothesis that junk DNA is sometimes generated in the RPA reaction. Figure 7 displays the data from the four sites, each having six water samples collected, including cell densities, shellfish toxicities, and the data generated by the RPA assay using the two different preparation techniques. No toxin data were generated for three samples due to the official control testing not being performed on these dates. On these instances, PST data from the closest available date was used, with these data points highlighted in red on Figure 7. The total PST concentrations observed in all sites were not only below the European maximum permitted limit (MPL) of 800 µgSTXeq/kg [54]. For this reason, only semi-quantitative data were generated for the samples, and, although this data were less accurate, they did give a reliable indication STX concentration in shellfish [55]. Moreover, the sampling sites experienced very mild *Alexandrium* blooms, with no requirements to close. This was well suited for assessing the performance of an early warning system, with each site having samples close to low trigger levels. The cell counts were for all *Alexandrium* species combined, with the trigger level in Scotland being 40 cells/L, equivalent to observing a single cell in a 25 mL sample, which when breached prompted further flesh tests. Two preparation techniques were compared for the RPA assay, the VolTRAX multiplexing kit and the RPB kit. For these two data sets, the number of reads aligning to any *Alexandrium* sp. Was displayed along with the number of reads aligning to specific species. The green filling on these indicated where a consensus sequence was generated that confirmed the presence of the organism both by BLAST analysis and a phylogenetic tree (phylogenetic trees and aligned consensus sequences available in Appendix A).

From a chronological perspective, the cell counts indicated the presence of *Alexandrium* sp. throughout the testing period, but there was a trend for toxicity in the latter half of the testing, except for site one, which had 22 µg STX eq/kg detected on the 15 June.

Sequencing of the RPA assay showed that the non-toxic species *A. tamutum* was present in all sites and most prevalent in earlier sampling dates, but this was confirmed as late as the 27 July [56].

*A. ostenfeldii* was present sporadically throughout the sampling period but only in one site. Although the species is associated with toxin production, local variants were associated with the production of spirolides rather than PSTs [57]. Yet, no correlation between these occurrences and levels of spirolides from shellfish flesh testing were found.

*A. tamarense*, now regarded as non-toxic, was confirmed at three of the sites from June through to September, but with a higher prevalence in the latter samples [9].

The PST producing *A. minutum* was observed with the lowest abundance of reads, and its presence was only confirmed in three samples using the RPB kit [58]. It was noted that the species presence was confirmed on each occasion, and levels of PST in shellfish were detected. Furthermore, although not confirmed by the generation of a consensus sequence, there were numerous reads aligning with the species from the site one sample that had levels of PST detected on the 15 June.

*A. catenella* is a toxic species (previously *A. tamarense* group I) formally referred to as *A. fundyense* and is thought to be the predominant toxic species in Scottish waters from historic chemotaxonomic data [8,9,53,59,60]. This species was confirmed at each site in the latter three sampling periods, with a notable correlation between its presence and the higher levels of PST being detected.

Although there was no expectation that the assay would be in any way quantitative, visual analysis of the data indicated there could be some correlations between the RPA product concentrations and cell counts. Due to the issues with obtaining accurate concentrations of the amplicon, the number of reads aligning with any *Alexandrium* references in the real-time alignment was used. The correlations observed using the RPB were not as good as when using the VMK; it is thought that this was primarily due to the manual levelling of barcodes before and after the PCR step. As such, these data were not reported. The Pearson’s correlation coefficient between microscopy cell count and total aligned reads using VMK resulted in r = 0.647 and *p* = 0.001; shellfish toxicity against VMK reads aligning with *A. catenella* were r = 0.802 and *p* ≤ 0.00001. However, no flesh toxin data were generated from site four on 08/09/2021, with data populated by a result gathered a week later, justifying its removal (correlations of r^2^ = 0.473 and *p* = 0.196 without removal of outlier). Comparison of cell counts and shellfish toxicities had poorer correlations of r = 0.395 with *p* = 0.561, which exceeded the critical level of 0.05, indicating that the data sets were statistically different (graphs showing this data can be found in Appendix A). This was due to the presence of the non-toxic species of *Alexandrium* in samples collected earlier in the year. By using the molecular data to screen the samples for the presence of toxic *A. catenella,* these data were significantly improved to r = 0.780 and r = 0.00001 (outlier removed). These data highlighted that the VolTRAX generated data were statistically similar to both the cell counts and shellfish toxicity. Furthermore, the VolTRAX data correlated better than cell counts when each datum was compared to toxicity.

However, the data generated by the early warning water sampling in the UK was used in a categorical manner rather than quantitative, being categorized as not detected, detected below trigger level (800 µg STX eq/kg), or detected above trigger level. In the instance of *Alexandrium*, and depending on the size of the Utermöhl chamber, the trigger level was activated as soon as cells were seen. In accordance with this, the data were converted to categorical format (positive or negative) by using the generation of a consensus sequence for *A. catenella* as an indication of a positive result for molecular data, counts above 20 cells/L for microscopy, and the presence of toxins in shellfish flesh above the analytical limit of detection. Once complete McNemar’s test was performed [61]. This identified no statistical difference between positive or negative shellfish toxicity results and the presence or absence of *A. catenella* reads using the RPB or VMK molecular approach tested, with *p* = 1 and *p* = 0.48, respectively. Conversely, cell count data by microscopy were statistically different to shellfish toxicity, (*p* = 0.046). By regarding samples having any species of *Alexandrium* confirmed as being positive, it was also possible to compare the molecular assays with the cell count. This highlighted no statistical difference, with both techniques resulting in *p* = 1.

The data generated indicated that the molecular assay described here had an improved limit of detection and accuracy in terms of identifying the toxin-causing variants. This would mean that a trigger level would have to be determined if such an approach were to be adopted, and, to achieve this, far more data would need to be generated to provide greater degrees of confidence in the results generated.

These findings represented a significant breakthrough and offered a viable alternative to light microscopy for an early warning tool for shellfish harvesting industries and/or competent authorities. Further development would greatly add to the value of the assay. The designing of a probe (or probes) to enable lateral flow would overcome the issues encountered for detecting amplification and could be used as a screen. Such a screen could be used to add value to light microscopy data, improving correlations with toxicity and preventing unnecessary additional sampling of shellfish flesh. However, longer term parallel analysis and testing of additional species will be critical prior to adoption for routine monitoring.

## 3. Materials and Methods

### 3.1. Biological Material and Sample Preparation

#### 3.1.1. Reference Materials

Algal cultures of *A. catenella* (1119/27), *A. tamarense* (1119/1), *A. minutum* (1119/48), *A. pacificum* (1119/52), and *A. tamutum* (1119/51) were acquired from the Culture Collection of Algae and Protozoa (CCAP, Oban, Scotland) and grown in 250 mL flask (75 cm^2^ growth area) containing L1 [62] medium at 17 °C and exposed to 14 h of light and 10 h of darkness per day.

#### 3.1.2. Sample Concentration

Two sample concentration techniques were compared: centrifugation and filtration. For centrifugation, 50 mL water samples were spun for 5 min at 8000 rpm before the supernatant was removed, and the pellet was resuspended in 400 µL of molecular-biology-grade water (MBG). The resuspension was pipetted into a 1.5 mL centrifuge tube, containing approximately ten 0.5 mm sterile glass beads, and submitted for cell lysis.

For filtration, 50 mL was pushed through a 25 mm filter holder (Cole-Palmer Item #: WW-29550-42), housing a 0.22 µm mixed cellulose ester membrane filter (Millipore, UK). The membrane filter was removed with tweezers and transferred to a 1.5 mL micro centrifuge tube, positioned so that the captured cells faced inwards to the tube and had 400 µL MBG water added for resuspension during lysis. To this tube, approximately ten 0.5 mm sterile glass beads were added.

#### 3.1.3. Performance Testing

Efficacy of each sample’s concentration protocols were tested by spiking 50 mL Lugol’s fixed sea water samples with 1, 10, 100, and 1000 cells of *Alexandrium minutum* for both methods in triplicate. Each method was also applied to a sample of the water that had no cells spiked and used as a negative control.

To see if Lugol’s affected the assay, 100 live *A. minutum* cells were spiked into fresh sea water and subjected to the centrifugation assay.

A negative control for both fixed and unfixed sea water samples was included in the study. The inclusion of a sample of 1000 *Alexandrium minutum* analysed in the absence of seawater represented a positive control for the study. An unfixed sea water sample was spiked with ~1000 live *A. minutum* cells to ensure that the assay worked comparably with fixed and unfixed samples.

#### 3.1.4. Cell Lysis

Cell lysis was achieved using a customized cordless oscillating multifunctional power tool (Einhell Varrito, EAN: 4006825618648). Fitted to the power tool was an adapted blade that held a 15 mL nylon centrifuge tube, into which lysis tubes were inserted and packed, so there was no movement independent of the outer tube (see Figure 8A,B). The device, referred onward as the “Osci-lyser”, delivered a rotational movement of 3.2°, and its adapted blade provided an 8.5 cm throw, resulting in approximately 5 mm movement on each oscillation. It had a variable speed selector, ranging from 183 to 333 Hz. The lowest speed was used for 20 s to achieve cell lysis in the study. After lysis, the sample was centrifuged for 1 min at 8000 rpm using the Bento lab before a 13.2 µL aliquot was taken for RPA amplification.

To test the efficacy of the Osci-lyser, a comparison was made against a bench top laboratory device, specifically, the MP biomedicals Fastprep-24 classic. This was achieved by performing triplicate DNA extractions on cultures of *Alexandrium*, approximately 50,000 cells in 400 µL of distilled water for each extraction. The fast prep was set to 4.0 M/S and run for 20 s. The Osci-lyser had the speed set to minimum, and a sample was lysed for 20 s. Once lysis was complete, samples were centrifuged, and 10 µL from each sample was quantified using a Qubit fluorometer.

### 3.2. RPA Assay

#### 3.2.1. Primer Design

Primers were designed manually using a nucleotide alignment of the D1 and D2 regions of the rRNA large subunit (LSU) of 47 *Alexandrium* reference sequences, with 24 species, using BioEdit [63]. Additionally, 10 other dinoflagellate genera were included to assess specificity. The aligned sequences and selected primers are available in Appendix A. Conserved regions of 36 bases were selected in D1 and D2 regions for the forward and reverse primers, respectively, to amplify a fragment with approximately 500 bp. Any polymorphism in position(s) of the annealing regions was accounted for with degeneracies. The size of the primers designed varied between 30 and 36 bases (Table 1); shorter primers were selected by removing from the end that resulted in the removal of degenerated positions [64]. Figure 1 shows location of primers in the rDNA cassette and provides an example alignment, including *Alexandrium.*

All possible combinations of the seven forward and the seven reverse candidate primers were tested in a single experiment. This consisted of 49 reactions using DNA extracted from a reference culture of *Alexandrium* spp. The experiment was repeated using the 8 primer pairs that yielded the best results to confirm which primer pair to use.

Once primers had been selected, they underwent in silico testing by BLASTn analysis against the NCBI data base [65]. This was performed on each primer, accounting for any variants in the binding region.

#### 3.2.2. RPA Reactions

RPA reactions were carried out using reagents provided by TwistDX (Cambridge, UK), which had two TwistAmp (TA) products that could generate sequence-able amplicons, the TA Basic and TA Liquid basic. All reaction volumes were 50 µL. The TA Basic was supplied as lyophilized pellets in individual 96 tubes, to which the primers, a rehydration buffer, and template were added. The TA Liquid Basic was supplied as a selection of reagents that must have been prepared into master mix. The TA Liquid Basic kit was used for preliminary testing associated with primer selection and the TA Basic for onward analysis of samples and method performance analysis. For each sample analysed, a 50 µL reactions was prepared in accordance with the relevant kit protocol and reactions performed using the Bento lab, set to 38 °C for 30 min. After amplification, samples were cleaned up using Agencourt AMPure XP magnetic beads, following their PCR amplicon protocol, using a 1:1 ratio of beads and reaction volume (50 µL). In brief, this involved: adding mag beads to the reaction mix, incubating the mix to allow DNA to attach to the beads, placing the tube on a magnet to attract the beads, removal of the spent reagents, two washes with 80% ethanol, and resuspension in either de-ionized water or, if it was to be analysed by the VolTRAX, in 10 mM Tris-HCl, pH 8.0.

#### 3.2.3. DNA Quantitation and Electrophoretic Analysis

To assess efficacy of RPA reactions, fluorometry and electrophoretic analyses were used. Fluorometry was performed on a Qubit 2.0 with a dsDNA HS Assay Kit (ThermoFisher, Dartford, UK), which was powered by the 18 V, 2 Ah Osci-lyser battery via a USB adapter and plug, which adjusted voltage from 5 V to 9 V, as shown in Figure 8C,D.

All gel electrophoresis was performed using 1% agarose gel at 110 V using a 100 bp ladder (Promega). This could have been conducted in the field using the Bento lab; however, all electrophoresis testing in the study was undertaken using laboratory equipment.

An Agilent Tapestation 4150 using a D1000 screen tape was used to confirm the sizes and quantities of the RPA products obtained.

#### 3.2.4. Specificity Testing

A mock community was generated to test assay specificity and potential for interference. It contained reference cultures of: *Skeletomena grethae* (CCAP1077/3), *Phaeodactylum tricornutum* (CCAP1052/1A), *Isochrysis galbana* (CCAP927/1), *Tetraselmis suecica* (CCAP66/4), *Prorocentrum lima* (CCAP1136/9), *Prorocentrum micans* (CCAP1136/25), *Scrippsiella acuminata* (CCAP1134/14), *Oxyrrhis marina* (CCAP1133/5), *Ansanella natalensis* (CCAP1132/1) *Kryptoperidinium foliaceum* (CAP1128/1, *Gymnodinium* sp. (CCAP1117/9), and *Ensiculifera imariensis* (CCAP1111/1). The resulting material was suspended in sterilized sea water, homogenized, and split into two, with one being spiked with approximately 100 cells of *Alexandrium minutum* (CCAP1119/48) and re-homogenized. The spiked and un-spiked mock communities were then subjected to the centrifugation method of concentration, cell lysis, and RPA reaction, sample clean up, and electrophoretic analysis. Any resulting amplicons were submitted for sequencing using the using the field sequencing kit.

In vitro inclusivity and exclusivity testing was then undertaken using reference cultures of both *Alexandrium* reference cultures: *Alexandrium catenella* (CCAP1119/6), *A. tamarense* (CCAP1119/9), *A. catenella* (CCAP1119/52), *A. minutum* (CCAP1119/48), *A. pacificum* (CCAP1119/52), and *A. tamarense* (CCAP1119/40) and three other species of dinoflagellates, specifically, *Lingulidinium polyedra* (CCAP1121/15), *Ansanella natalensis* (CCAP1132/1), and *Ostreopsis ovata* were also tested.

Each culture was subjected to the centrifugation method of concentration, cell lysis, and RPA reaction, sample clean up, and electrophoretic analysis. Any resulting amplicons were submitted for sequencing using the using the rapid PCR barcoding library kit.

### 3.3. Library Preparation and Nanopore Sequencing

#### 3.3.1. Library Preparation Tools

All thermal incubation and centrifugation requirements associated with ONT library preparation were fulfilled using the Bento lab portable molecular laboratory station at full speed for 1 min. The Bento Lab was also used for the PCR required when multiplexing using the rapid PCR barcode (RPB).

The VolTRAX V2b (V2b) was used to automate sample preparation and for multiplexing. It was a USB powered, multi-purpose, microfluidics device designed to automate sample preparation for nanopore sequencing platforms (image available in Appendix A). It had the potential to perform PCR amplification, mag bead clean up, and fluorometric quantitation. The multiplexing kit used in this study allowed for simultaneous analysis of 10 samples and did not use PCR.

#### 3.3.2. Sequencing Device

All sequencing was performed on a MinION MK1b connected to custom-made, field-deployable computer referred to as the XavION, shown in Figure 9. The exact specifications of the XavION are available in the Appendix A, but, in brief, the unit was built around the Nvidia Jetson Xavier AGX 32gb, running Linux. The XavION took inspiration from a system developed by Benton Miles but took advantage of more advanced computer hardware [66]. To allow portability and field deployment, the computer was mounted in an IP67-rated case, connected to a touch screen monitor, and had a USB hub to connect low-power-consuming ancillaries, such as a keyboard and mouse. Installed on the rear of the case was an IP67-rated electrical socket, allowing the 185Wh internal battery to be charged in situ while retaining the IP rating (when closed). The data storage was expanded via an M2 SSD drive and had both Bluetooth and Wi-Fi connectivity. A transparent plastic divider was installed to separate the computer hardware from the MinION, with holes to facilitate cooling and interaction with both computer and battery control buttons. An Elegoo Mars 2 pro, a 3D resin printer, was used to make custom mounts to hold the MinION and computer module in place position and provide a tube rack with 1 position, having a magnet for mag bead clean up. External power was delivered by the Ecoflow River Max providing 576 Wh capacity.

#### 3.3.3. Field Sequencing Library Preparation Kit

The ONT field sequencing kit ref: SQK-LRK001 (LRK) was used for the analysis of single samples. The kit consisted of three lyophilized reagents and a single rehydration buffer, requiring long term storage at 4 °C but could withstand short periods at room temperature. The kit was used in accordance with the appropriate ONT protocol.

#### 3.3.4. Rapid PCR Barcode Library Preparation Kit

The ONT rapid PCR barcoding kit ref: SQK-RPB004 (RPB) was used for multiplexing up to 12 samples and performing a rapid library preparation. The protocol was adapted slightly to be suitable for the short fragments being analysed by reducing the PCR method from 6 min, stated in the protocol, to 30 s. After PCR barcoding, the Qubit was used to quantify each barcode and calculate the volume of each sample to reach similar concentrations for each.

#### 3.3.5. VolTRAX Multiplexing Library Preparation Kit

Automated barcoding and library preparation was tested using the VolTRAX multiplexing Kit ref: VSK-VMK004 (VMK) in conjunction with the ONT VolTRAX V2b sample preparation using a VolTRAX cartridge ref: VCT-V2002D. The kit was used in accordance with the appropriate ONT protocol and could multiplex 10 samples into a single library ready for sequencing.

### 3.4. Case Study: Environmental Samples

The environmental study made use of legacy material collected in 2021 as part of the routine monitoring program by Scottish Association of Marine Science, Oban, Scotland on behalf of Food Standards Scotland. Lugol’s fixed samples had been analysed using light microscopy for *Alexandrium* cells, with counts being performed on 50-mL samples using the Utermöhl method [23]. In total, 24 samples were used, originating from four shellfish harvesting sites and collected over a 6-month period (May to September). Sub samples of the water used for the cell counts were shipped to Cefas laboratory and had DNA extracted for a previous study, which was undertaken using the Qiagen DNAeasy Power biofilm kit (QIAGEN, Hilden, Germany) [29]. These DNA extracts were submitted for RPA amplification in duplicate, with one having further sample prep undertaken using the rapid PCR barcoding kit and the other using the VolTRAX multiplexing kit. These had to be performed on different RPA amplicons due to them requiring preparation in different elution buffers after mag bead clean up. Sequencing was performed on the XavION using r 9.4.1 flow cells, employing the high accuracy base calling model (Guppy 6.4.6).

In addition, the study incorporated shellfish flesh toxicity data generated for the routine monitoring program using high performance liquid chromatography with fluorometric detection (HPLC-FLD) [67]. As only samples exceeding a reporting limit of 400 µgSTXeq/kg were submitted for the full quantitation, semi-quantitative data were used. Although not as accurate, the assay had been proven to provide a good indication of overall toxicity [55]. For three of the samples, no flesh toxicity data were generated since levels of lipophilic toxins had breached the action limit in previous tests, meaning the sites had been closed, mitigating the need for PST data collection. For these sites, the most recent available data were used. In both instances, data were available from within a week of the water sample being collected.

### 3.5. Bioinformatic and Statistical Analysis

#### 3.5.1. Bioinformatics

MinKNOW 5.4.3 was used to align sequence data against a list of 50 reference sequences of *Alexandrium* (Appendix A).

For each sample being analysed, consensus sequences were generated. These were generated from reads passing a quality threshold of Q7 and used NGSpeciesID version 0.2.1 [68]. The following command was used to achieve this:
“NGSpeciesID - m 400 --s 200 --mapped_threshold 0.8 --aligned_threshold 0.8 --rc_identity_threshold 0.8 --abundance_ratio 0.01 --ont --consensus --t 20 --fastq input.fastq --outfolder output_folder --medaka --medaka_model r941_min_high_g360”

The command stipulated that amplicon lengths of 400 ± 200 bp were selected, mapping and alignment thresholds were set to 0.8 for clustering of sequences, rc identity threshold was set to 0.9 to remove duplicated consensus sequences, and abundance ratio was set to 0.01. The Medaka polishing was set to use the r941_min_high_g360 model. The number of computer cores used could be determined and, in the instance below 20, were selected by the --t 20 function.

Once generated, the consensus sequences were submitted for BLAST alignment against the NCBI reference database [69]. Sequences that aligned with *Alexandrium* references > 98%ID were then submitted for phylogenetic assessment by iqtree version 2.1.2 [70], using a reference list inspired by Tomoyuki Shikata et al. and available in the Appendix A [43].

#### 3.5.2. Statistical Analysis

Pearson’s correlation coefficient was used to assess the data generated from the environmental sample case study, with r and *p* values reported (critical value = 0.05). These were used to compare sequencing data in the form of reads aligning to either any species of *Alexandrium* and cell counts from the microscopy data set and reads aligning to *A. catenella* and shellfish toxicity. Additionally, molecular assay was used as a screen to determine if *A. catenella* was present, and only cell counts used on such instances were used to see if they correlated with the toxicity data.

To analyse categorial data, all shellfish toxicities, cell counts, and molecular data were assigned a positive or negative result. Statistical interactions between approaches were calculated by performing McNemar’s tests. Analysis was performed in Rstudio (version 1.4.1717) and utilized the “tidyverse” and “stats” packages. A data set was deemed positive if the *p* value exceeded the common significance level of 0.05.

## Figures and Tables

**Figure 1 toxins-15-00372-f001:**
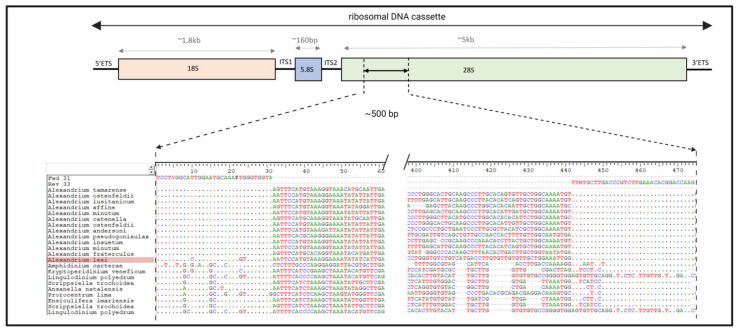
Graphical representation of ribosomal DNA cassette, location targeted by primers and example alignment of primer regions. Positions in the alignment matching the primers are represented by a dot and variants are shown as letters.

**Figure 2 toxins-15-00372-f002:**
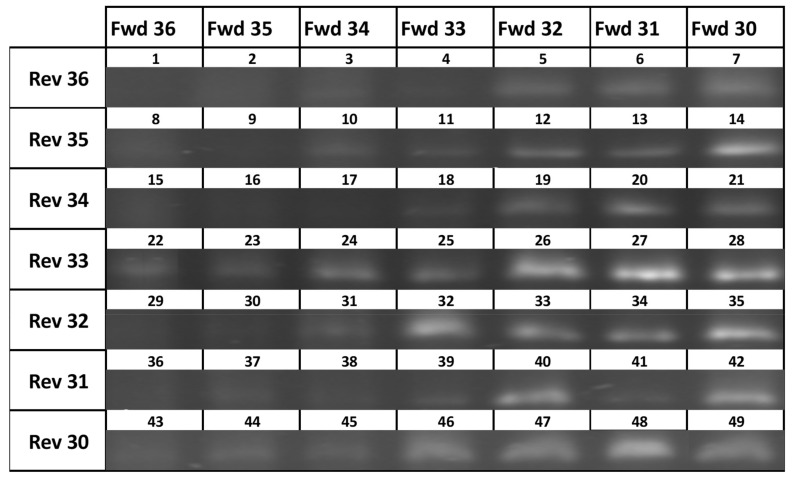
Primer performance matrix. Numbers refer to reaction number with gel electrophoresis bands at ~500 bp. (original Gel available in Appendix A).

**Figure 3 toxins-15-00372-f003:**
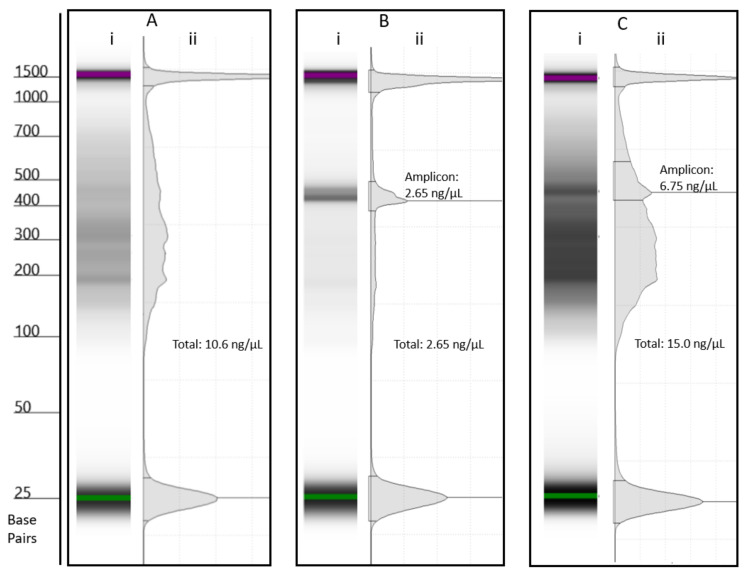
Environmental samples analysed by capillary electrophoresis showing a (**i**) gel and (**ii**) chromatogram for: (**A**) a negative sample with an interference peak from non-specific amplification, (**B**) a positive sample with no interference peak, and (**C**) a sample experiencing very low levels of amplification and a significant level of interference. Note: the peaks at 25 and 1500 bp are reference peaks. Concentrations for the amplicon and total concentration are shown.

**Figure 4 toxins-15-00372-f004:**
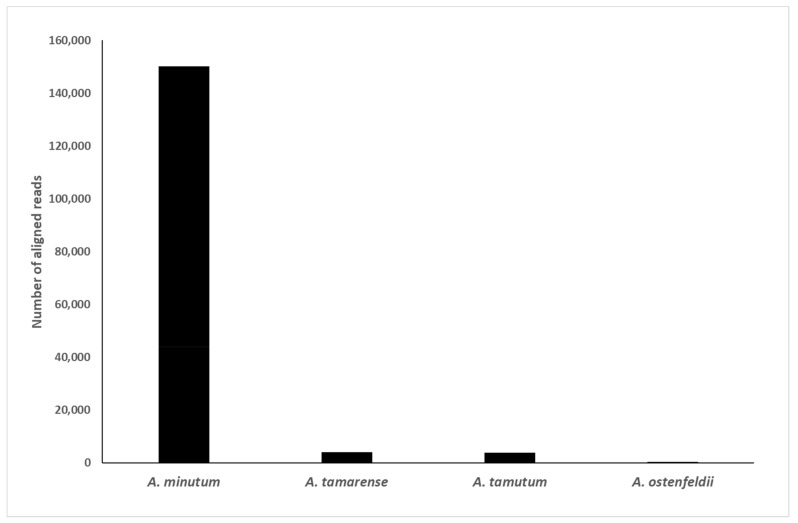
Sequences from the mock community sample spiked with *Alexandrium minutum* cells were aligned against references for *Alexandrium* spp. using MinKNOW 5.4.3.

**Figure 5 toxins-15-00372-f005:**
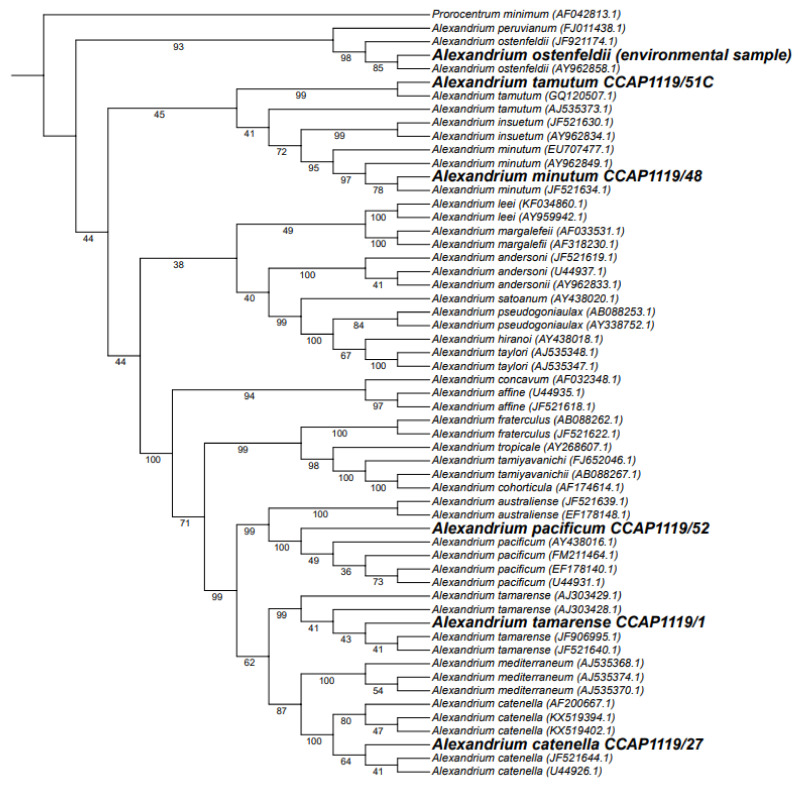
Phylogenetic tree using a maximum likelihood approach and 10,000 bootstraps with branch lengths not shown (a tree including branch lengths is included in the Appendix A). Sequences in bold were generated in this study.

**Figure 6 toxins-15-00372-f006:**
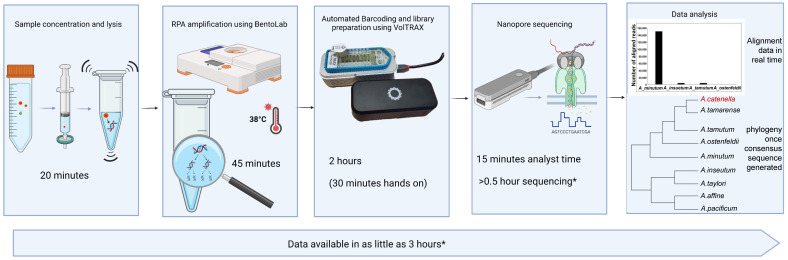
Chronological workflow of sample preparation for VolTRAX barcoding and library preparation. * The time required for sequencing will be variable, primarily dependent on the pores available on the flow cell.

**Figure 7 toxins-15-00372-f007:**
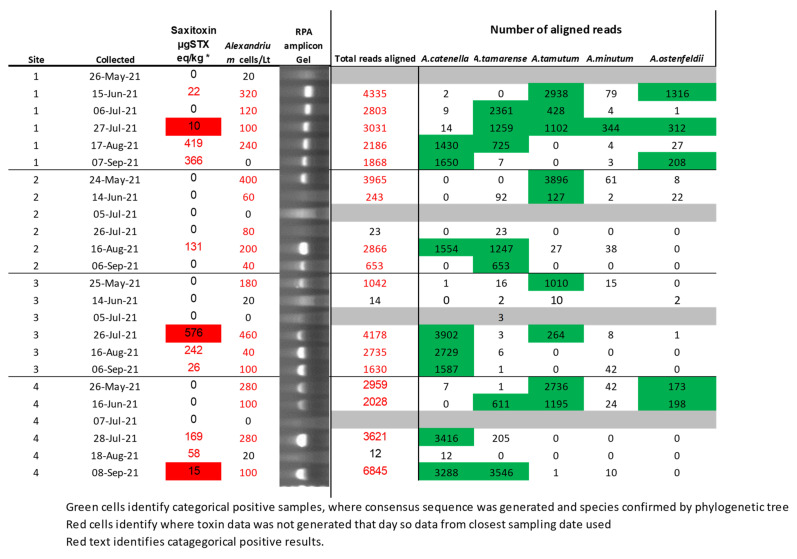
Environmental sample data from four sites experiencing sub action limit toxin events (<800 µgSTXeq/kg). * Shellfish flesh saxitoxin concentration result obtained by semi-quantitative HPLC-Fld. Red cells identify where toxin data were not generated that day; thus, data from closest sampling data were used.

**Figure 8 toxins-15-00372-f008:**
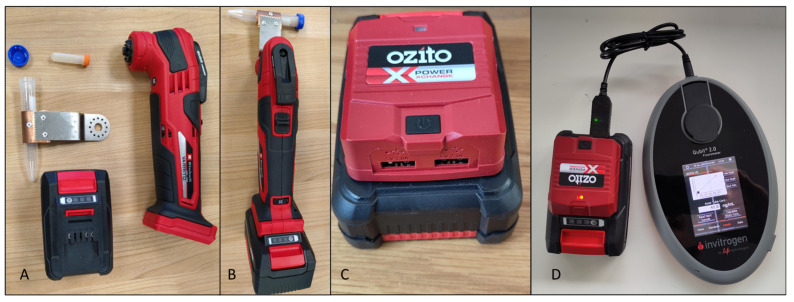
Field deployable sample preparation tools: (**A**) component parts of Osci-lyser, (**B**) the Osci-lyser assembled and ready to use, (**C**) USB adaptor for battery, providing 1.0- and 2.1-amp outputs, (**D**) Battery powering Qubit 2.0.

**Figure 9 toxins-15-00372-f009:**
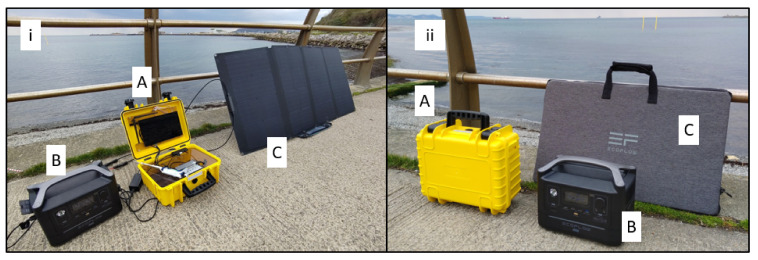
Field deployable sequencing device: (**A**) XavION sequencing computer, (**B**) External Power supply, (**C**) 200-watt solar panel. (**i**) deployed for use, (**ii**) packed for transport.

**Table 1 toxins-15-00372-t001:** List of primers tested with selected primers highlighted * and degeneracies underlined.

	**Forward Primers:**
	Fwd36: 5’ CACAYTCCTYGGCATTGGAATGCAAAKTGGGTGGTA 3’
	Fwd35: 5’ ACAYTCCTYGGCATTGGAATGCAAAKTGGGTGGTA 3’
	Fwd34: 5’ CAYTCCTYGGCATTGGAATGCAAAKTGGGTGGTA 3’
	Fwd33: 5’ AYTCCTYGGCATTGGAATGCAAAKTGGGTGGTA 3’
	Fwd32: 5’ YTCCTYGGCATTGGAATGCAAAKTGGGTGGTA 3’
*	**Fwd31: 5’ TCCTYGGCATTGGAATGCAAAKTGGGTGGTA 3’**
	Fwd30: 5’ CCTYGGCATTGGAATGCAAAKTGGGTGGTA 3’
	**Reverse Primers:**
	Rev36: 5’ CTTGGTCCGTGTTTCAAGACGGGTCAAGCAKAADCA 3’
	Rev35: 5’ CTTGGTCCGTGTTTCAAGACGGGTCAAGCAKAADC 3’
	Rev34: 5’ CTTGGTCCGTGTTTCAAGACGGGTCAAGCAKAAD 3’
*	**Rev33: 5’ CTTGGTCCGTGTTTCAAGACGGGTCAAGCAKAA 3’**
	Rev32: 5’ CTTGGTCCGTGTTTCAAGACGGGTCAAGCAKA 3’
	Rev31: 5’ CTTGGTCCGTGTTTCAAGACGGGTCAAGCAK 3’
	Rev30: 5’ CTTGGTCCGTGTTTCAAGACGGGTCAAGCA 3’

**Table 2 toxins-15-00372-t002:** A summary of different experimental protocols tested, along with the results. Successful and failed amplifications are identified by “+” and “−”, respectively.

Experiment	Concentration Method	Mag Bead Clean Up	*Alexandrium* Cells Spiked	Amplification		
				A	B	C
**Sensitivity testing**	Centrefugation	yes	1000	+	+	+
		yes	100	+	+	+
		yes	10	+	+	+
		yes	1	+	+	−
		yes	0	−	−	−
	Filtration	yes	1000	+	+	+
		yes	100	+	+	+
		yes	10	+	+	+
		yes	1	+	+	+
		yes	0	−	−	−
	None *	yes	1	+	+	+
		yes	0	−	−	−
**Live Cells**	Centrifugation	yes	100	+	+	+
**Mock community**	Centrifugation	yes	100	+		
		yes	0	−		

* Cells added directly to 400 µL lysis solution to ensure no loss of material.

## Data Availability

Not applicable.

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
