# Peer review of "Combining Nanopore Sequencing with Recombinase Polymerase Amplification Enables Identification of Dinoflagellates from the Alexandrium Genus, Providing a Rapid, Field Deployable Tool"

_toxins, 2023, doi:10.3390/toxins15060372_

Round 1

Reviewer 1 Report

A very interesting research/approach! I think that the present manuscript could be published prior to some minor revision, since I do not have any major concerns.  It is a well-written manuscript and  the authors are analytical and the tables that they are presenting indeed help the author very much. It is conscience and within the scope of the Journal. 

I would recommend the authors to carefully read again the manuscript and correct some mistakes regarding the articles referenced. Some references do not match with what the authors report. Also, some linguistic/grammatical mistakes could be corrected .

 some linguistic/grammatical mistakes could be corrected

Author Response

Thank you for taking the time to review the manuscript, it is very pleasing to hear you found the article interesting. We have taken some time to go through the manuscript and made some minor grammatical changes and updated the references.

Reviewer 2 Report

This well-designed study adds relevant information to the molecular identification of Alexandrium-contaminated samples at a species level. The only concern I have is related to the results in Figure 1. It is not clear the source of DNA for these reactions. Is it from one single species? The combination of DNA from several species? Please clarify this minor issue.

The manuscript is written with a very well usage of English.

Author Response

Thank you for taking the time to review the manuscript, we are very pleased with the review you submitted. In response to your comments, we have specified that the matrix shown in figure 1 was generated using a reference culture of Alexandrium catenella.

Reviewer 3 Report

The article is devoted to the mpdern sequencing methods development in the case of seatype Dinoflagellate Alexandrium. The method described in the article is of high reserach interest and would be interesting for readers. The research desigh is appropriate and described in the small details that shows an article scientifically ready for publishing.

There are few points must be improved: 

Figure 3 - please, provoide any tipy of mathematical statistics for the data presented.

Introduction - please provide more information for readers why the Dinoflagellate Alexandrium is of the high reserahc interest and nanopore sequencing methods required for this type. Which kinds of Dinoflagellate Alexandrium could be toxic for humans and why it is important to make quick express sequence for this type.

Imporve English, especially, 2.2.1 sensitivity test

English is understandable, but minor corrections must be done all over the text

Author Response

Thank you for taking the time to review the manuscript, we were very pleased with your review, especially that you felt the the manuscript is of “high research interest”. In response to your comments:

  • For figure 3, the data generated used MinKNOW, the oxford nanopore software. However, to give more detail on the underpinning statistical tools used by MinKnow to we have highlighted that MiniMap2 is used as an imbedded piece of software.
  • We have added some additional detail to the introduction to address the points you made.
  • We have reviewed the manuscript, making minor changes and rewritten section 2.2.1.

Reviewer 4 Report

The authors identified species of the genus Alexandrium by a double-stranded enzyme polymerase amplification method and nanopore sequencing of a 500 bp fragment of the resulting rRNA large subunit. The analytical sensitivity and specificity of the assay was assessed using seawater samples spiked with different Alexandrium species. In addition, the McNemar paired test on qualitative data highlighted no statistical difference between samples identified as positive or negative for Alexandrium by phylogenetic analysis and the presence or absence of toxins in shellfish. The overall idea is correct. Where necessary, the author is adding some research to enhance the value of the paper, its impact on humans and its future.

In 2.1.1 primer design, the reasons for the selection of primers need to be developed and analysed to avoid the use of uncertain terms such as possible, suspected.

Missing from subheading 2.2.2.

2.4 The next level heading is labelled incorrectly.

2.5 Things to note during sampling. volTRAX data correlate better than cell counts compared to toxicity and it is recommended that this can be shown visually using graphs.

5.1.2 Transfer of the resuspension to a centrifuge tube, how to transfer, instruments required for the transfer, notes.

Instrument conditions should be indicated for centrifugation, cell lysis and RPA reactions, sample clean-up and electrophoretic analysis of spiked and unspiked mock communities.

Author Response

Thank you for taking the time to review the manuscript and providing the valuable guidance on how to improve it. In response to these comments, we have:

  • Reworded section 2.1.1.
  • Rectified the subheadings in section 2.
  • Provided a graph to show the improved correlation which will be included in supplementary material.
  • I have highlighted in the text that we transferred the liquid using a pipette.
  • We have included details on the instrumental conditions where appropriate.

Round 2

Reviewer 4 Report

All the comments were revised and I think this work could be accepted in the current version.